# Detection of Cyclic Imine Toxins in Dietary Supplements of Green Lipped Mussels (*Perna canaliculus*) and in Shellfish *Mytilus chilensis*

**DOI:** 10.3390/toxins12100613

**Published:** 2020-09-24

**Authors:** Paz Otero, Carmen Vale, Andrea Boente-Juncal, Celia Costas, M. Carmen Louzao, Luis M. Botana

**Affiliations:** Departamento de Farmacología, Farmacia y Tecnología Farmacéutica, Facultad de Veterinaria, Universidad de Santiago de Compostela, 27002 Lugo, Spain; mdelcarmen.vale@usc.es (C.V.); andrea.boente.juncal@usc.es (A.B.-J.); celia.costas.sanchez@usc.es (C.C.); mcarmen.louzao@usc.es (M.C.L.); luis.botana@usc.es (L.M.B.)

**Keywords:** 13-desmethyl spirolide C, pinnatoxin-G, dietary supplements, *Perna canaliculus*, *Mytilus Chilensis*, UPLC-MS/MS

## Abstract

Seafood represents a significant part of the human staple diet. In the recent years, the identification of emerging lipophilic marine toxins has increased, leading to the potential for consumers to be intoxicated by these toxins. In the present work, we investigate the presence of lipophilic marine toxins (both regulated and emerging) in commercial seafood products from non-European locations, including mussels *Mytilus chilensis* from Chile, clams *Tawerea gayi* and *Metetrix lyrate* from the Southeast Pacific and Vietnam, and food supplements based on mussels formulations of *Perna canaliculus* from New Zealand. All these products were purchased from European Union markets and they were analyzed by UPLC-MS/MS. Results showed the presence of the emerging pinnatoxin-G in mussels *Mytilus chilensis* at levels up to 5.2 µg/kg and azaspiracid-2 and pectenotoxin-2 in clams *Tawera gayi* up to 4.33 µg/kg and 10.88 µg/kg, respectively. This study confirms the presence of pinnatoxins in Chile, one of the major mussel producers worldwide. Chromatograms showed the presence of 13-desmethyl spirolide C in dietary supplements in the range of 33.2–97.9 µg/kg after an extraction with water and methanol from 0.39 g of the green lipped mussels powder. As far as we know, this constitutes the first time that an emerging cyclic imine toxin in dietary supplements is reported. Identifying new matrix, locations, and understanding emerging toxin distribution area are important for preventing the risks of spreading and contamination linked to these compounds.

## 1. Introduction

Lipophilic marine toxins in mollusc constitute an important threat to human health and high number of intoxications occur every year [1]. The legislated group of lipophilic marine toxins includes four different chemical groups: yessotoxins (YTXs) azaspiracids (AZAs), pectenotoxins (PTXs), and okadaic acid (OA) and its derivatives, the dinophysistoxins (DTXs). Only few compounds of each group are regulated: yessotoxin (YTX), homo-yessotoxin (Homo-YTX), 45-hydroxy-yessotoxin (45-OH-YTX), 45-hydroxy-homo-yessotoxin (45-OH-homo-YTX), azaspiracid-1 (AZA-1), azaspiracid-2 (AZA-2), azaspiracid-3 (AZA-3), pectenotoxin-1 (PTX-1), pectenotoxin-2 (PTX-2), OA, dinophysistoxin-1 (DTX-1), dinophysistoxin-2 (DTX-2), and dinophysistoxin-3 (DTX-3). When LC-MS methodology is used, the concentration in a sample has to be referred to a predominant toxin of the group, named the reference compound (RC) and using the toxicity equivalency factors (TEFs) values for the other analogues from the same group [2,3]. The RCs for lipophilic toxins are YTX, AZA-1, PTX-2, and OA. The use of TEFs requires the knowledge of the toxicity of each analogue present in a sample to link analytical data into their toxicity [2,3]. Several countries have proposed limits in molluscs for some lipophilic toxins, for example, in the European Union (EU), levels in shellfish for human consumption have to be below 3.75 mg eq YTX/kg, 0.16 mg eq AZA, and 0.16 mg eq OA/Kg (for the OA and PTX toxin group) [4,5]. Until 2013, all official determinations were performed through in vivo assays, but after that year, the analysis by LC-MS/MS were progressively increasing, according to the legislation 15/2011 [6].

In the past decade, evidence has grown for the occurrence of non-regulated toxins, spirolides (SPXs) and pinnatoxins (PnTXs), in mollusc from EU waters [7,8,9,10,11,12,13,14]. These compounds belong to the cyclic imine (CIs) group and chemistry, they are macrocyclic molecules with imine and spiro-linked ether moieties [15,16]. Their mode of action is related to the interaction with muscle-type and neuronal nicotinic acetylcholine (ACh) receptors (nAChR) [9]. No incident of human intoxication has been attributed to CI so far and currently there are no regulations in molluscs in EU or in other regions worldwide. However, EFSA has recommended a maximum level of 400 µg sum of SPXs/kg shellfish meat [17].

The high occurrence of CIs leads us to consider the existence of these toxins in mollusc from non-EU locations and in seafood formulations commercialized as dietary products, so that all these products could constitute a new risk for consumers since the demand of food ingredients from marine environment has increased in the latest years [18]. Seafood as a dietary component has many human health benefits and many are connected to the consumption of important omega-3 fatty acids, but also those food supplements based on mussels´ formulations could have phycotoxins. The most common marine food supplements are based on fish (hake, trout, tuna, etc.,), freshwater microalgae (*Haematococcus pluvialis*, *Chlorella vulgaris,* etc.,), macroalgae (*Fucus vesiculosus*, *Cystoseira osmundace*), cyanobacteria (*Arthrospira platensis, Arthrospira maxima,* etc.,), and molluscs (*Perna canaliculus*) and they are marketed in dose form including capsules, tablets, or pills [19,20,21]. Marine food supplements are legislated as foods and thus, the supplier who place the product on the market has the responsibility for guaranteeing the safety of these products [22]. In Europe, although benefits of food supplements have to be supported by scientific evidence and the EFSA approval, they are not subjected to pharmacovigilance as with medicines and they do not require the same quality and homogeneity among batches [23]. For this reason and because of the increase of emerging toxin detection worldwide, it is convenient to study the contamination level of lipophilic toxins in marine dietary products.

In this study, we investigate the presence of lipophilic marine toxins in both, molluscs from non-EU locations (New Zealand, Chile, SouthEast Pacific, and Vietnam) but commercialized in Europe and marine food supplements. The aim is to know the toxin profile, their levels, and if they constitute a new risk to human health. Despite CI toxin group do not pose an official method for its detection, EFSA recommends the development and optimization of LC-MS/MS methods for the analyses of CIs. Figure 1 shows the structure of the main compounds representatives of each toxin group analyzed in this study, YTX, AZA-1, PTX-2, OA, 13-desmethyl spirolide C (SPX-13) and pinnatoxin G (PnTX-G).

## 2. Results

### 2.1. Analysis of Lipophilic Marine Toxins in Molluscs

The availability of certified reference standard PnTX-G has allowed it to be included in the EU-Harmonized Standard Operating Procedure (SOP) together with SPXs and the other toxins currently legislated. Figure 2A shows the chromatogram of the standard PnTX-G (C_42_H_63_NO_7_) at a concentration of 12.5 ng/mL in methanol. Mass spectrometry detection was operated in positive mode and the product ion spectra of [M + H]^+^ ions obtained for *m*/*z* 694.4 at three different collision energies (CE) are shown in Figure 2B (CE 40 eV), Figure 2C (CE 54 eV), and Figure 2D (CE 60 eV). Fragmentation pathways are in concordance with those previously reported [11,13,24] in which the *m*/*z* 164.1 fragment ion (C_11_H_18_N^+^) is also common for some SPXs, including SPX-13, 13,19-didesmethyl spirolide C (SPX-13,19), spirolide C (SPX-C), spirolide-D (SPX-D), spirolide H (SPX-H), and spirolide I (SPX-I). Analysis also showed fragmentation clusters at *m*/*z* 458.3 and *m*/*z* 676.3 (Figure 2D).

Molluscs, *Mytilus chilensis* (*M. chilensis*), *Tawera gayi (T. gayi)* and *Meretrix lyrate (M. lyrate)* were analyzed with the EU-Harmonized Standard Operating Procedure (SOP) for the determination of regulated lipophilic toxins (OA, DTX-1, DTX-2, YTX, 45 OH-YTX, HomoYTX, 45 OH-HomoYTX, PTX-1, PTX-2, AZA-1, AZA-2, and AZA-3) [25] and including SPX-13, 13,19-didesmethyl spirolide C (SPX-13,19), 20-methyl spirolide C (SPX-20G) and PnTX-G. The MS transitions used in the multiple reaction monitoring (MRM) method for each toxin are indicated in Material and method section. Figure 3 shows the chromatograms from the analysis of lipophilic marine toxins. Results evidenced the presence of PnTX-G (RT = 3 min) in all samples from brand A (Figure 3A) and brand B (Figure 3B) of *M. chilensis*. PnTXs, produced by *Vulcanodinium rugosum* dinoflagellate, have been detected in seafood from the Mediterranean Sea in 2013 and 2018 [10,24] and in the Atlantic coast of Spain last year [11,12,13]. These results confirm also the presence of these toxins in mussels from Chile. Chromatograms from *T. gayi* shows the presence of AZA-2 and PTX-2 (Figure 3C). The limits of detection (LODs) were 0.1 µg/kg for OA, DTX-1, DTX-2, PTX-1, and PTX-2; 0.3 µg/kg for AZA-1, AZA-2, and AZA-3; 1.2 µg/kg for YTXs; and 0.1 µg/kg for SPX-13, SPX-13,19, and PnTX-G. The limits of quantification (LOQs) were: 0.3 μg/kg (OA, DTXs and PTXs), 0.9 μg/kg (AZAs), 3.6 μg/kg (YTXs), 0.3 μg/kg (SPXs), and 0.4 μg/kg (PnTX-G). Table 1 collects the quantification for each toxin present in the samples upper the LOQs. Levels were low, up to 4.00 µg/kg, 4.33 µg/kg, and 10.88 µg/kg for PnTX-G, AZA-2, and PTX-2, respectively, and no other toxin was detected under LOQs. The signal suppression/ enhancement (SSE) value because of the matrix was 94.2%, 40.5%, and 65.0% for PTX-2, AZA-2, and PnTX-G, respectively. For three toxins, there was a negative effect which entailed a suppression of signal. To express the content of each toxin group as µg AZA equivalent (eq)/kg or µg PTX eq/kg, the individual content of AZA-2 and PTX-2 was multiplied by the TEFs provided by EFSA which are 1.8 and 1 for AZA-2 and PTX-2, respectively.

### 2.2. Analysis of Lipophilic Marine Toxins in Dietary Supplements

Afterwards, dietary supplements were analyzed by UPLC-MS/MS. Toxins were extracted with two solvents, methanol (extraction 1) and methanol:water (extraction 2) explained in Material and methods section. Using both extraction methods, results showed the existence of the emerging toxin SPX-13 in 6 out of 9 samples analyzed after both extractions and no other lipophilic toxins were found. Comparing extraction methods, better results were obtained for the extraction 2 in which the percentage of water employed was the same than naturally is present in fresh mussels, which is 82% [26]. SPX-13 recovery for the extraction 2 was 98.3% ± 1.4. Figure 4 shows the chromatograms of the food supplements of *Perna canaliculus* (*P. canaliculus*) where the emerging toxin SPX-13 was identified in brand D (Figure 4A) and brand F (Figure 4B). These chromatograms correspond to samples extracted with method 2 and they were compared with the standard (Figure 4C), whose ion ratios were reproducible. From our knowledge, this is the first report of spirolides in mussel-based food supplements. SPX-13 originated by the marine dinoflagellate *Alexandrium*
*ostenfeldii* or *Alexandrium peruvianum* [27,28] and is the most common analogue of SPXs.

After identification, SPX-13 was quantified in all samples and amounts are shown in Table 2. Levels ranged from 33.2 to 97.9 µg/kg. Quantification was determined by an external standard calibration and considering the matrix effect which were calculated using matrix in the absence of toxins as a blank sample. Linearity was assessed by calibration curves (nine points) prepared in both methanol and matrix within the range of 0.1 ng/mL to 25 ng/mL, obtaining a proper adjusted linear regression in both cases (r^2^ ≥ 0.998) (Figure 5). The SSE value due to the matrix was 97.5%, so there was a negative effect which entailed a suppression of signal (−2.5%).

Because the CIs consist of a large number of molecules, mainly without commercially available standards, and considering the occurrence of different analogues of PnTXs worldwide and SPX-13 in the food supplements, we also considered the possible existence of other CIs different from those first analyzed in the samples. LC-MS/MS methods are based on a targeted screening that seeks to find a list of known compounds, while missing other unknown toxic compounds that could be present in the sample. Moreover, it seems probable that PnTX-G is the precursor of PnTX-A, PnTX-B, and PnTX-C due to metabolic and hydrolytic transformations in molluscs [29]. LC-MS/MS methods are based on a targeted screening that seeks to find a list of known compounds, while missing other unknown toxic compounds that could be present in the sample. Thus, dietary supplements and molluscs were re-analyzed using a MS method including transitions for a wide range of CIs for which standards are not available. Transitions are included in material and methods section (Table 3). The screening of CIs by UPLC-MS/MS showed that no other analogues were in the samples.

## 3. Discussion

This study shows the first evidence of the newly discovered toxin PnTX-G in mussels *M. chilensis* and SPX-13 in dietary supplements of mussels *P. canaliculus.* PnTXs were first detected in viscera of Japanese *Pinna muricata* in 1995 [30] and then, they expanded to other areas and species. These toxins were found in Pacific oysters (*Crassostrea gigas*) and razor fish (*Pinna bicolor*) from South Australia and Northland New Zealand [31,32], mussels from Norway [7], Canada [33], and clams (*Venerupis decussata*) from France [24]. Recently, PnTXs were also detected in mussels (*M. galloprovincialis*) from the Atlantic Coast of Spain [11,12,13] and in other countries from the Atlantic and Mediterranean coastline including Scotland, Northern Ireland, Ireland, Italy, Norway, and Portugal [34]. Concentrations of PnTXs in shellfish are generally low, however, PnTX-G was found to reach the concentration of 1200 µg/kg in mussels collected in the coast of France [24,29]. Lamas et al. (2019) found that the maximum PnTX-G levels in the South of Europe are during winter [11]. However, Arnich et al. (2020) showed that PnTX-G peaks were observed between June and September from molluscs collected from the French Mediterranean coast between 2013 and 2017 [14]. Last year, because of the occurrence of PnTXs in France, the French Agency for Food, Environmental and Occupational Health and Safety has developed a scientific opinion regarding the presence of PnTXs in shellfish, setting an acceptable contamination value of 23 μg PnTX-G/kg (total meat). Thus, a possible risk for human consumers occurs when bivalves accumulate toxins higher than this value as a result of ingesting the toxic microalgae [14]. In the present study, PnTX-G reached a maximum of 5.25 µg/kg (*M. chilensis*), much lower than the proposed level. A very recent study has allowed to stablish an oral LD_50_ for PnTx-G of 208 μg/kg and a provisional No Observed Effect Level (NOEL) of 120 μg/kg [35].

Clams *T. gayi* showed AZA-2 and PTX-2 in small quantities. No trace levels neither AZA-1 nor AZA-3 were detected. These results agree with others in which AZA-2 was the most abundant or unique analog among the three (AZA-1, AZA-2, AZA-3) monitored [12,36]. In the present study, clams *T. gayi* from the SouthEast Pacific exhibited up to 4.33 µg/kg of AZA-2, levels comparable to those found in mussels from the Atlantic coast of Spain (1.8–3 µg/kg) [12,36]. Although AZA-1 was the predominant analogue in Chilean mussels, AZA2 predominated in Pacific scallop (*Argopecten purpuratus*) from the same country [37]. From a safety point of view, toxicology evaluations of AZA-2 indicate this analog is less toxic than AZA-1 [38]. The estimation of AZAs lowest observed adverse effect level (LOAEL) was established at 1.9 µg AZA eq/kg body weight (b.w.) [38] and the acute reference dose (ARfD) was set at 0.2 µg AZA eq/kg b.w. [38]. Currently, the estimation of AZA-related shellfish toxicity is based on the quantification of AZA1–3 [1], and the TEFs extrapolated from mouse i.p. LD_50_ of these three compounds, with values of 1.0, 1.8, and 1.4 for AZA-1, AZA-2, and AZA-3, respectively [38]. EFSA identified 400 g of shellfish meat as the high portion size to be used in the acute risk assessment of marine biotoxins [39]. It was noted that consumption of a 400 g portion of shellfish meat containing AZAs at the current EU limit of 160 μg AZA eq/kg shellfish meat would result in a dietary exposure of 64 μg AZA eqs. For a 60 kg adult this is approximately 1 μg AZA eq/kg b.w. In order for a 60 kg adult to not exceed the ARfD, a 400 g portion of shellfish should not contain more than 12 μg AZA eq, i.e., 30 μg AZA eq/kg shellfish meat [39], much higher than the amount found in the present study. In 2018, more proper TEFs from oral LD_50_ results (in mice) were proposed, resulting in 1.0, 0.7, and 0.5 for AZA-1, AZA-2, and AZA-3 respectively [40]. These TEFs seems more adequate since considering the potential human exposure by oral route, AZAs TEFs should be calculated by comparative oral toxicity data [40]. Considering these new TEFs, the amount of total AZA in *T. gayi* is lower than 3 µg eq AZA/kg.

Similarly, PTX-2 levels in *T. gayi* (4.41–10.88 µg/kg) were also low compared to other Pacific bivalves (82.0 µg/kg) [41]. The toxicological database for PTX-group toxins is limited and comprises mostly studies on their acute toxicity in mice. There are no reports on adverse effects in humans associated with PTX-group toxins. PTX-2 was acutely toxic to mice by i.p. injection (LD_50_ = 219 µg/kg) and it was not overtly toxic to mice by the oral route at doses up to 5000 µg/kg. [42]. EFSA had proposed a TEF value of 1 for PTX-1 and PTX-2 [43]. Although the oral toxicity is not well defined, it was stablished a LOAEL of 250 μg/kg b.w. and an ARfD of 0.8 μg PTX eq/kg b.w. Thus, for a 60 kg adult to avoid exceeding the ARfD of 0.8 μg PTX eq/kg b.w., a 400 g portion of shellfish should not contain more than 48 μg PTX eq corresponding to 120 μg PTX-2 eq/kg shellfish meat [43]. In Asian molluscs, several combinations of lipophilic marine toxins were reported, including OA/YTX, OA/PTX-2, YTX/OA, PTX-2/OA, PTX-2/GYM, GYM/PTX-2 [44], and also AZA-2/PTX-2 [45]. It appears that OA is the most often recorded lipophilic toxin in mixtures, as well as the predominant toxin in the mixture [44]. Toxicological studies reporting the effects of the co-association AZA-2/PTX-2 were not found in the literature.

SPXs were discovered in Canadian mussels in the early 1990s [46] and nowadays are distributed worldwide, not only in edible species (clams, cockles), but also in other species which have a significant role in the food-chain, such as gastropods (*Gibbula. umbilicalis*, *Nucella. lapillus*, *Patella. intermedia, Monodonta* sp.) and starfish (*Marthasterias. glacialis)* [47]. It was not recorded, neither the presence of SPXs in dietary supplements nor in *P. canalicus* mussels before. Marine food supplements constitute an important part of the global market and are produced from different sources that provide a multitude of bioactive molecules such us proteins, unsaturated essential fatty acids, vitamins, minerals, carotenoids, polysaccharides [18,20,48]. Particularly, green lipped mussels’ powder is known for their therapeutic use in the rheumatoid or osteoarthritis treatment [49] with significant anti-inflammatory activity because of the content of omega-3 polyunsaturated fatty acids [50]. This dietary food supplement is marketed globally, easily available and announced as beneficial for health. Nevertheless, dietary products could also lead to potential toxic risks related to the contamination by toxins [51]. This is the first report of an emerging marine CI toxin found in mussel-based supplements. However, a variety of other toxic substances has been recorded in marketed samples of food supplements considering both scientific literature and Rapid Alert System for Food and Feed notifications [51]. These contaminants include metals, polychlorinated biphenyls (PCBs), mycotoxins, and cyanotoxins [51]. There is no information regarding the presence of CIs in fresh *P. canaliculus.* Instead, it was reported the presence of YTXs (16–32 μg/kg) and PTX-2 (2–15 μg/kg) [52]. SPX-13 in food supplements were at levels up to 97.9 µg/kg and the content of 500 mg of concentrated green mussel was provided in each capsule. It was viable to design a proper extraction procedure with 97% of toxin recovery and able to quantify SPX-13 in the range of 13.4–427.35 µg/kg with almost inexistent matrix effect.

The availability of blank matrix is an important issue when developing detection methods by LC-MS/MS since majority of molluscs already have marine toxins naturally. This could be solved, introducing internal standards that behave similarly to the analogues of interest analyzed. Also, current MS/MS detection methodology for marine toxins are based on a defined target list of toxins and are not usually appropriate for the identification of unknown compounds [53]. To guarantee the consumers’ safety, food quality assurance would have to detect the presence of toxic unknown compounds which include newly discovered toxins, the identification of known toxins in areas and species where they had not been recorded before. When assessing the associated risk for human health because of the geographical expansion of marine toxins, there is an urgent need for robust analytical methodologies that can detect a wide range of known or emerging toxins in different matrix.

## 4. Materials and Methods

### 4.1. Molluscs and Dietary Supplements Acquisition and Sample Preparation

A total of 21 marine food products were purchased during November and December 2019. Product type, origin, and amount obtained from the market are described in Table 3. They were from four locations (New Zealand, Chile, SouthEast Pacific and Vietnam) and belonged to six commercial brands (called A, B, C, D, E, and F). Samples 1–6 are frozen mussels (*M. chilensis*) and samples 6–12 are frozen clams (*T. gayi* and *M. lyrate*), all purchased in a local market in Lugo (Spain). Samples 13–21 are green lipped mussels powder (*P. canaliculus*) and they were obtained by three different EU distributors. Samples from the same brand come from the same batch and species (and locations) were chosen at random, since they are frequent products available in local markets.

Once in the laboratory meat molluscs were defrosted, removed from shell and homogenized (Ultra Turrax^TM^, Staufen, Germany), stored in bags at −20 °C, and preserved from oxygen and light. Each sample was a homogenate of the tissue of 15 individual mollusc (100 g approx.). Food supplement capsules were opened, and the powder content was passed to a 45 mL empty tube. Each sample was a homogenate of 40 capsules content (20 g approx.)

### 4.2. Chemicals

Acetonitrile, methanol, sodium hydroxide, and hydrochloric acid were obtained from Panreac (Barcelona, Spain). Formic acid was purchased from Merck (Darmstadt, Germany) and ammonium formate was from Fluka (Sigma-Aldrich, Madrid, Spain). All solvents were of HPLC or analytical grade and water was obtained from a water purification system (Milli-Q, Millipore, Madrid, Spain). Certified reference materials were provided by Cifga (Lugo, Spain): dinophysistoxin-1 (DTX-1 8.08 ± 0.41 µg/g), dinophysistoxin-2 (DTX-2 2.54 ± 0.14 µg/g), okadaic acid (OA 24.92 ± 1.82 µg/g), tessotoxin (YTX 7.42 ± 0.49 μg/g), 1a-homoyessotoxin (homo-YTX 7.68 ± 0.44 μg/g), azaspiracid-1 (AZA-1 1.36 ± 0.07 μg/g), azaspiracid-2 (AZA-2 1.33 ± 0.11 μg/g), and azaspiracid-3 (AZA-3 1.30 ± 0.09 μg/g), 13-desmethyl spirolide C (SPX-13, 7.29 ± 0.36 µg/mL), 13,19-didesmethyl spirolide C (SPX-13,19, 7.51 ± 0.38 µg/mL), and 20-methyl spirolide G (SPX-20, 7.01 ± 0.61 µg/mL. Certified calibration solutions for pinnatoxin-G (PnTX-G, 2.43 ± 0.11 μg/g) and for pectenotoxin-2 (PTX-2, 5.58 ± 0.16 μg/g) were purchased from The Institute for Marine Biosciences, National Research Council, Halifax, NS, Canada.

### 4.3. Toxin Extraction

Mollusc were extracted following the EU-Harmonized Standard Operating Procedure (SOP) for the determination of lipophilic marine biotoxins in molluscs by LC-MS/MS [25]. All bivalves were extracted according the procedure from Section 6.2 of the SOP; the Annex C was not applied for cooked samples. The amount of 2.00 g of tissue homogenate were weighed into a centrifuge tube. Then, 9 mL of methanol was added to it and the sample was homogenized by vortex mixing during 3 min. Afterwards, the samples were centrifuged (3700 rpm × 10 min) at 20 °C and the supernatant was transferred to a 20-mL volumetric flask. The extraction of the residual tissue pellet was repeated with another 9 mL of methanol using a high-speed homogenizer (T25 digital Ultra-Turrax, IKA^®^-Werke GmbH & Co. KG, Staufen, Germany). After centrifugation (3700 rpm × 10 min) at 20 °C, the supernatants were combined into a final volume of 20 mL with methanol. The volume of 15 mL was concentrated to 5 mL to improve the sensibility of the analyses. One aliquot of 500 µL was filtered through 0.22 µm filter and then analyzed by LC-MS/MS. Food supplements were extracted adding methanol to the powder samples (extraction 1) or using methanol after a rehydration of the food supplements (extraction 2). For the extraction 1, 2 g of powder food supplement homogenate were weighed into a centrifuge tube and extracted with 9 mL of methanol. The extract was homogenized via vortex and then, the samples were centrifuged (3700 rpm × 10 min) and the content was transferred to 20 mL volumetric flash. The extraction of the residual tissue pellet was repeated with another 9 mL of methanol. After centrifugation (3700 rpm × 10 min), the supernatants were combined into a final volume of 20 mL with methanol. The extraction 2 was performed in two steps. First, an amount of 1.64 mL of water was added to 0.36 g of power food supplement in order to have a humid mussel extract with the same water content as the fresh mollusc (82 ± 0.02 g H_2_O/100 g [26]). After that, it was extracted according to extraction 1.

### 4.4. UPLC-MS Analysis

Analyses were performed by a 1290 Infinity ultra-high-performance liquid chromatography system coupled to an Agilent G6460C Triple Quadrupole mass spectrometer equipped with an Agilent Jet Stream ESI source (Agilent Technologies, Waldbronn, Germany). The toxins were separated using a column AQUITY UPLC BEH C18 (2.1 × 100 mm, 1.7 µm, Waters, Barcelona, Spain) at 40 °C. Mobile phase A was 100% water and B acetonitrile-water (95:5), both containing 50 mM formic acid and 2 mM ammonium formate. The gradient program with a flow rate of 0.4 mL/min was started with 30% B and then a linear gradient to 70% B in 3 min. After an isocratic hold time linear of 1.5 min at 70% B and return to the starting conditions of 30% B in 0.1 min. Finally, 30% B was kept for 1.99 min before the next injection. The samples in the autosampler were cooled to 4 °C and the injection volume was 5 µL. Source conditions were: 350 °C of drying gas temperature with 8 L/min flow, nebulizer gas pressure of 45 psi (Nitrocraft NCLC/MS from Air Liquid, Madrid, Spain), sheath gas temperature of 400 °C and a flow of 11 L/min. The capillary voltage was set to 4000 V in negative mode with a nozzle voltage of 0 V and 3500 V in positive mode with a nozzle voltage of 500 V. Analyses were performed in multiple reaction monitoring (MRM) acquisition mode, selecting two transitions per molecule. The first transition of each toxin indicated below was used for quantification and second one used as qualifier. Transitions for regulated toxins were: 45-OH-homo-YTX (*m*/*z* 1171.5 > 1091.5, *m*/*z* 1171.5 > 869.5), 45-OH-YTX (*m*/*z* 1157.5 > 1077.5, *m*/*z* 1171.5 > 871.5), Homo-YTX (*m*/*z* 1155.5 > 1075.5, *m*/*z* 1155.5 > 869.4), YTX (*m*/*z* 1141.5 > 1061.5, *m*/*z* 1141.5 > 855.4), PTX-1 (*m*/*z* 892.5 > 821.5, *m*/*z* 892.5 > 213.2), PTX-2 (*m*/*z* 876.5 > 823.5, *m*/*z* 892.5 > 213.2), AZA-1 (*m*/*z* 842.5 > 824.5, *m*/*z* 842.5 > 806.5), AZA-2 (*m*/*z* 856.5 > 838.5, *m*/*z* 856.5 > 820.5), AZA-3 (*m*/*z* 828.5 > 810.5, *m*/*z* 828.5 > 792.5), OA/DTX-2 (*m*/*z* 803.5 > 255.1, *m*/*z* 803.5 > 113.2), and DTX-1 (*m*/*z* 817.5 > 255.1, *m*/*z* 817.5 > 113). Collision energy of each fragment are described in [12]. The MS/MS method for the screening of emerging toxins are described in Table 4. The MS/MS operated in positive ionization mode and the cell acceleration voltage (CAV) was 4 volts. Dwell was 6 for all toxins.

### 4.5. Method Performance, Recovery, and Matrix Correction

Analytical method performance assessment was performed for three days following the EU-Harmonized SOP for Lipophilic toxins [25] and the guidelines proposed by the Regulation (EC) 657/2002 [54]. The external standard calibration curves were made in methanol with nine levels in the range of 0.09 ng/mL to 25 ng/mL. Linearity was assessed by the regression coefficients of the quantification curves which had to be bigger than 0.99 and the slope variation between the sets of the calibration curve which had to be lower than 25% to be considered as acceptable. Sensitivity of the method was evaluated with the slope of the calibration curves and with the limit of detection (LOD) and limit of quantification (LOQ) which were calculated based on an signal-to-noise ratio of 3 and 10, respectively, using triplicate injections (*n* = 3) of standard solutions with quantities near the limits.

To address the effects of the extraction procedure and of the matrix, spiked mollusc extracts and dietary supplements (free of toxins) were used as a blank for toxin recovery and matrix correction. To calculate the signal suppression/ enhancement (SSE) factor due to the matrix, blank samples were extracted following extraction procedures described above. Nine concentration levels of each standard were diluted in toxin-free extracts (blank sample) and then it was analyzed by LC-MS/MS. The slopes of the curves were employed for calculating the SSE value according to the following equation: SSE (%) = 100 × (Slope of spiked extract curve/Slope of standards curve in solvent). If SSE value is 100%, no matrix effect is observed whereas if the value is higher than 100%, a positive matrix effect due to an enhancement of the ionization is observed. If this value is below 100% there is a negative effect, which entails a suppression of the signal due to ionic suppression. The corrected concentration considering the recovery and matrix effects was calculated as follows:(1)$$\frac{\text{µ}g\;\mathrm{toxin}}{\mathrm{kg}}=\left(\frac{\text{µ}g}{\mathrm{kg}}\right)\mathrm{EXTERNAL}\;\mathrm{CALIBRATION}\;\times \;\left(\frac{100}{\%R\;\mathrm{spiked}\;\mathrm{extract}}\right)\;\times \;\left(\frac{100}{\%\mathrm{SSE}}\right)$$
where: (µg/kg) EXTERNAL CALIBRATION is the concentration calculated by external calibration prepared in methanol with 9 levels in the range of 0.09 to 25 ng/mL.

## Figures and Tables

**Figure 1 toxins-12-00613-f001:**
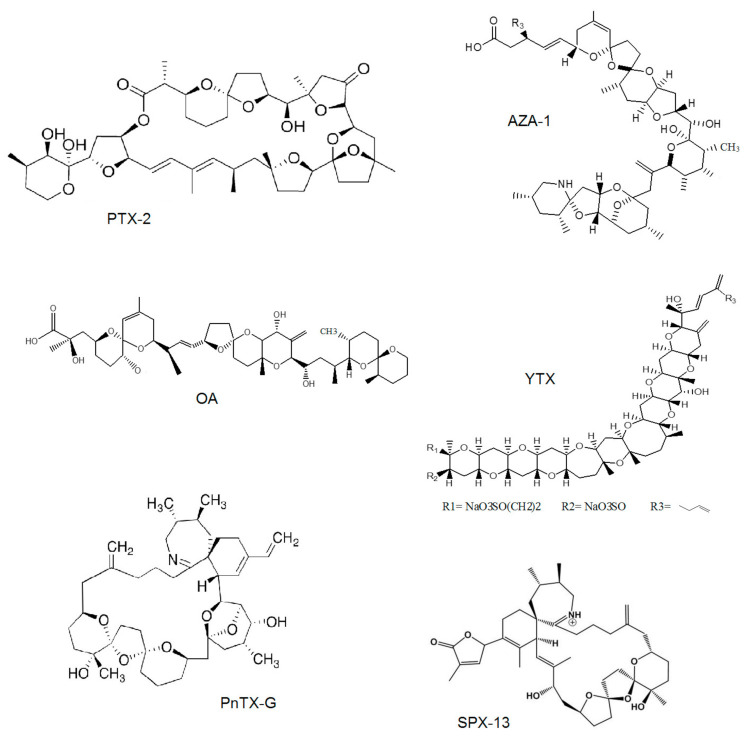
Structure of pectenotoxin-2 (PTX-2), azaspiracid-1 (AZA-1), okadaic acid (OA), yessotoxin (YTX), 13-desmethyl spirolide C (SPX-13), and pinnatoxin-G (PnTX-G).

**Figure 2 toxins-12-00613-f002:**
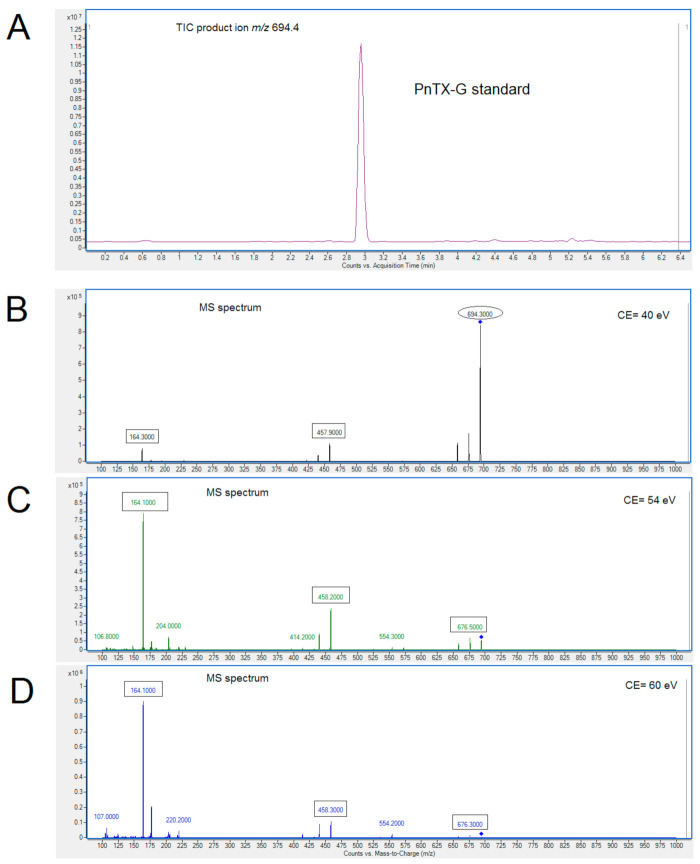
Chromatogram (**A**) and MS spectrum of PnTX-G standard at collision energy (CE) of 40 eV (**B**), 54 eV (**C**) and 60 eV (**D**). Standard concentration: 12.5 ng/mL.

**Figure 3 toxins-12-00613-f003:**
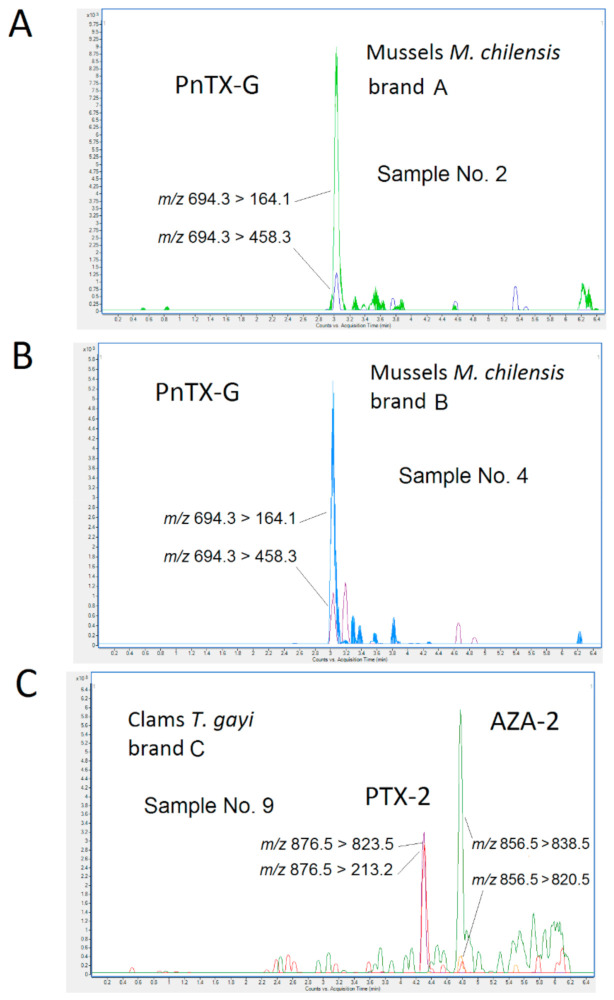
Multiple reaction monitoring (MRM) chromatograms of *Mytilus chilensis* (*M. chilensis*) brand A (**A**) and band B (**B**) and *Tawera gayi* brand C (**C**).

**Figure 4 toxins-12-00613-f004:**
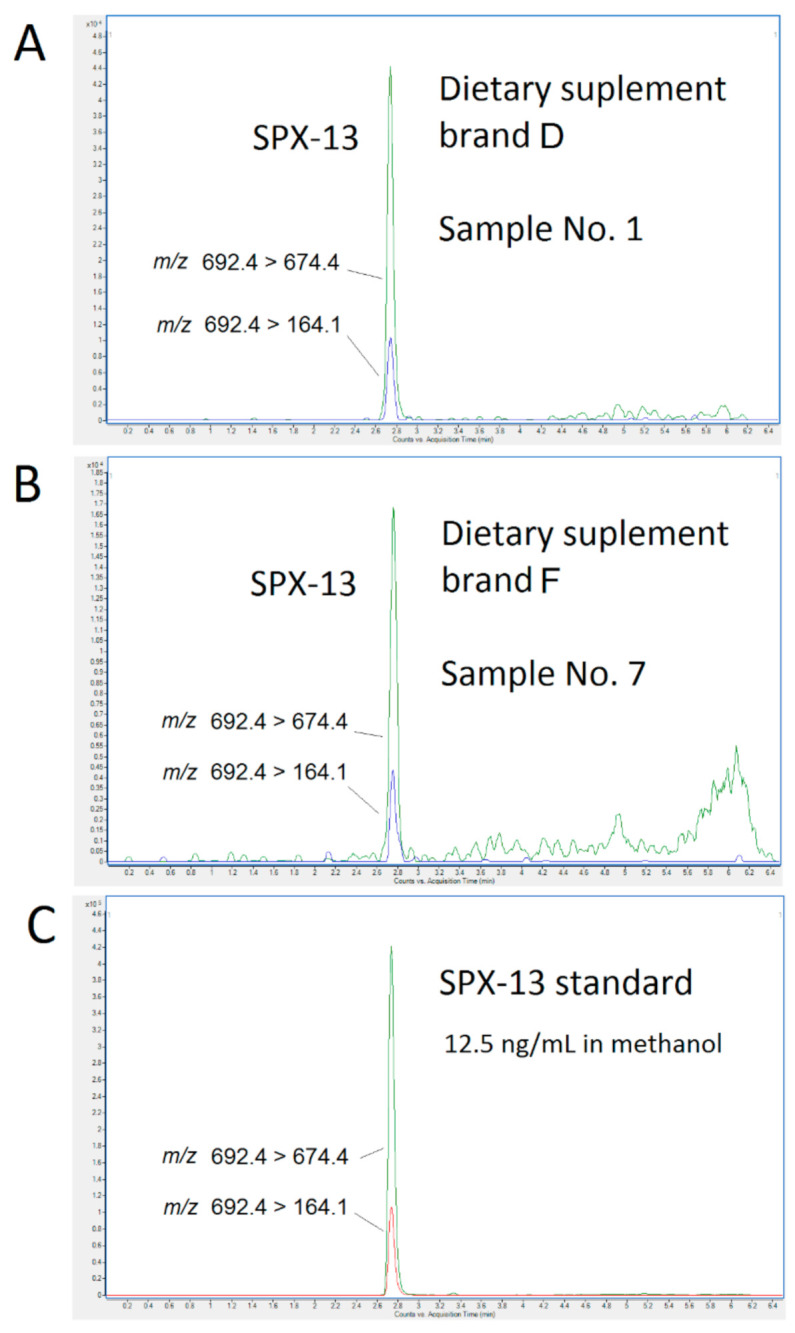
Multiple reaction monitoring (MRM) chromatograms of green lipped mussel powder (*Perna canaliculus*) brand D (**A**) and band F (**B**) and standard of 13-desmethyl spirolide C (SPX-13) (**C**).

**Figure 5 toxins-12-00613-f005:**
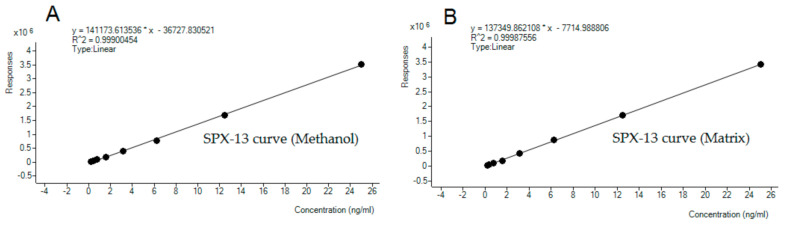
Calibration curves for spirolide 13-desmethyl spirolide C (SPX-13) in methanol (**A**) and in matrix (the powder from the dietary supplements of green lipped mussels) (**B**) in the range 0.19–25 ng/mL.

**Table 1 toxins-12-00613-t001:** Concentration of PnTX-G in *Mytilus chilensis* mussels from Chile and AZA-2 and PTX-2 in *Tawera gayi* clams from Southeast Pacific.

No.	Species	Brand	AZA-2 (µg/kg)	µg AZA eq/kg	PTX-2 (µg/kg)	µg PTX eq/kg	PnTX-G (µg/kg)
1	*Mytilus chilensis*	A	~	~	~	~	2.9
2	*Mytilus chilensis*	A	~	~	~	~	5.2
3	*Mytilus chilensis*	A	~	~	~	~	4
4	*Mytilus chilensis*	B	~	~	~	~	3.1
5	*Mytilus chilensis*	B	~	~	~	~	3
6	*Mytilus chilensis*	B	~	~	~	~	3.2
7	*Tawera gayi*	C	4.09	7.36	4.41	4.41	~
8	*Tawera gayi*	C	4.33	7.79	10.88	10.88	~
9	*Tawera gayi*	C	4.25	7.65	5.84	5.84	~
10	*Meretrix lyrata*	B	~	~	~	~	~
11	*Meretrix lyrata*	B	~	~	~	~	~
12	*Meretrix lyrata*	B	~	~	~	~	~

The term ~ means below the limit of quantitation (LOQ). LOQ (PTX-2) = 0.3μg/kg, LOQ (AZA-2) = 0.9 μg/kg and LOQ (PnTX-G) = 0.4 μg/kg.

**Table 2 toxins-12-00613-t002:** Concentration of SPX-13 in the food supplements, green lipped mussels’ power.

No.	Species	Brand	SPX-13 (µg/kg Dry Product)
13	*Perna canaliculus*	D	97.9
14	*Perna canaliculus*	D	90.86
15	*Perna canaliculus*	D	85.6
16	*Perna canaliculus*	E	<LOQ
17	*Perna canaliculus*	E	<LOQ
18	*Perna canaliculus*	E	<LOQ
19	*Perna canaliculus*	F	39.14
20	*Perna canaliculus*	F	33.2
21	*Perna canaliculus*	F	33.5

Limit of quantitation (LOQ, SPX-13) = 13.4 µg/kg.

**Table 3 toxins-12-00613-t003:** Food samples and origin. Amount obtained are also included.

No.	Food Product	Species	Origin	Brand	Sample Amount
1	Cooked Mussels	*Mytilus chilensis*	Chile	A	360 g
2	Cooked Mussels	*Mytilus chilensis*	Chile	A	360 g
3	Cooked Mussels	*Mytilus chilensis*	Chile	A	360 g
4	Cooked Mussels	*Mytilus chilensis*	Chile	B	275 g
5	Cooked Mussels	*Mytilus chilensis*	Chile	B	275 g
6	Cooked Mussels	*Mytilus chilensis*	Chile	B	275 g
7	clams	*Tawera gayi*	SouthEast Pacific	C	600 g
8	clams	*Tawera gayi*	SouthEast Pacific	C	600 g
9	clams	*Tawera gayi*	SouthEast Pacific	C	600 g
10	clams	*Meretrix lyrata*	Vietnam	B	1000 g
11	clams	*Meretrix lyrata*	Vietnam	B	1000 g
12	clams	*Meretrix lyrata*	Vietnam	B	1000 g
13	Green lipped mussels powder	*Perna canaliculus*	New Zealand	D	101 g
14	Green lipped mussels powder	*Perna canaliculus*	New Zealand	D	101 g
15	Green lipped mussels powder	*Perna canaliculus*	New Zealand	D	101 g
16	Green lipped mussels powder	*Perna canaliculus*	No information	E	500 mg
17	Green lipped mussels powder	*Perna canaliculus*	No information	E	500 mg
18	Green lipped mussels powder	*Perna canaliculus*	No information	E	500 mg
19	Green lipped mussels powder	*Perna canaliculus*	No information	F	45 g
20	Green lipped mussels powder	*Perna canaliculus*	No information	F	45 g
21	Green lipped mussels powder	*Perna canaliculus*	No information	F	45 g

**Table 4 toxins-12-00613-t004:** MS/MS method parameters for non-regulated marine toxins. Precursor and product ions monitored (*m*/*z*), collision energy (CE) and fragmentor voltage (Frag) are included.

Toxin	Precursor Ion [M + H]^+^	Product Ion	CE	Frag
Pinnatoxin A	712.4	458.3; 164.1	54; 60	112
Pinnatoxin-B/C	741.5	458.3; 164.1	54; 60	112
Pinnatoxin D	782.5	446.3; 164.1	54; 60	112
Pinnatoxin E	784.5	446.3; 164.1	54; 60	112
Pinnatoxin F	766.5	446.3; 164.1	54; 60	112
Pinnatoxin G	694.5	458.3; 164.1	54; 60	112
Gymnodimine A	508.3	392.4; 490.4	54; 40	180
Gymnodimine B/C	524.4	488.4; 506.4	40; 54	180
Gymnodimine D	524.4	346.4	45	180
12-methyl Gymnodimine A	522.4	406.4; 504.4	54; 40	180
13-desmethyl spirolide C	692.4	674.4;164.1	30; 54	180
13,19-didesmethyl spirolide C	678.4	660.4; 164.1	30; 54	149
20-methyl spirolide C	706.47	688.4; 164.1	30; 54	152
Spirolide A	692.5	150.1	70	180
Spirolide B	694.5	150.1	70	180
Spirolide C	706.5	458.3; 164.1	42; 54	180
Spirolide D	708.5	458.3; 164.1	42; 54	180
Spirolide E	710.5	444.3	60	180
Spirolide F	712.5	444.3	60	180
Spirolide G	692.5	378.3; 164.1	42; 54	180
Spirolide H	650.4	402.3; 164.1	42; 54	180
Spirolide I	652.5	402.3; 164.1	42; 54	180
27-hydroxy-13,19-didesmethyl SPX-C	694.4	464.3; 180.1	42; 54	180
27-hydroxy-13-desmethyl SPX-C	708.4	478.4; 180.1	42; 54	180
27-oxo-13,19-didesmethyl SPX-C	692.4	444.3; 178.1	42; 54	180

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
