# Peer review of "Detection of Cyclic Imine Toxins in Dietary Supplements of Green Lipped Mussels (Perna canaliculus) and in Shellfish Mytilus chilensis"

_toxins, 2020, doi:10.3390/toxins12100613_

Round 1

Reviewer 1 Report

This is the first report presenting occurrence data of 13-desmethyl spirolide C in dietary supplements and PnTX-G in Chilean mussels. This study is original and interesting even though the number of samples in the research design is limited.  

The lack of clarity and the numerous imprecisions affect the comprehension of the study . The article is therefore rambling and vague at some points (see comments below), and consequently needs substantive and form improvements.

Comment n°1:

Title: I suggest reformulating the title in a way to globalize the two experiments highlightening detection of the Cis; the current title looks like a juxtaposition of two completely distinct experiments (food supplements and shellfish).

Comment n°2:

L27, L41, L114, L119, L178, and L268: the use of the term “emerging toxins” is excessive throughout the article. In several cases, “non regulated toxins” or “newly discovered” seems more adequate. Most of the toxins were recently discovered and were not really monitored so far; therefore, it is difficult to affirm that those toxins really emerging in specific areas/locations.

Comment n°3:

L27: « first detection » is more appropriate than « first description ».

Comment n°4:

L3: several words are in italics. Is there a reason?

Comment n°5:

L34: define the meaning of the term: “The reference compound”.

Comment n°6:

L35: please indicate where those levels shall be respected? In Europe, worldwide? Could you also add regulation references?

Comment n°7:

L37: “official toxicity determination” à “official determination” seems more appropriate.

Comment n°8:

L39: please correct the sentence. The LC-MS/MS method could be applied since 2011 according to the legislation 15/2011.

Comment n°9:

L48: it should be noted at some part of the article that no human intoxication linked to the exposition to CIs have been reported, so far (unless new data are available).

Comment n°10:

L49: the term ”occurrence” is more appropriate than recurrence all over the article.

Comment n°11:

L49: CIs instead of CI.

Comment n°12:

L64: the statement: ”the increase of emerging toxins worldwide” needs some explanation. Is it effectively an increase or due to methodology improvements and better knowledge?

Comment n°13:

L74: “pectenotoxin-2” and not “pentotoxin-2”

Comment n°14:

L73 - Figure 1: should be introduced at the end of the paragraph after it citation.

Comment n°15:

L75 - § Results: you do not talk about the results below the LOD and/or LOQ for the different toxins studied. Therefore, it is difficult to understand which toxins were really analyzed. Please add some sentences in this chapter to clarify the paper.

Comment n°16:

L78 and L92: please, replace “LC-MS reference method” by “EU-Harmonized Standard Operating Procedure (SOP)” as in L228.

Comment n°17:

L79 to L87: please, move this paragraph in the material and method part.

Comment n°18:

L94: YTX and not YTXs

Comment n°19:

L80: “PnTX-G (C42H63NO7) eluted in the minute 3 and the chromatographic peak show a high intensity of 1.2 x 107 cps “. This information are not useful.

Comment n°20:

Figure 2, 3 and 4: sizes shall be reduced.

Comment n°21:

L91: “mollusks” here, whereas “molluscs” is written elsewhere. Please harmonize.

Comment n°22:

L92: replace “for the determination of lipophilic toxins” by “for the determination of regulated lipophilic toxins”

Comment n°23:

L94: for SPX-13 and PnTX-G the full name is not used whereas for the two others spirolides you mention the full name and the abbreviation.

Comment n°24:

L96: you used in two lines “shown”, “shows”, “showed”. Please avoid these repetitions.

Comment n°25:

L97-98: there is a lack of coherence between the text and Figure 3 and Table 1 as regards the brands of M. Chilensis. In the main text, you speak about brand D and E but in Figure 1 and Table 1 there is only Brand A and B.

Comment n°26:

L 98: « Besides » ?

Comment n°27:

L 100: “Table 1 shows the quantification” is not a complete sentence

Comment n°28:

L 101: can you mention the years instead of “recently”? It will be more accurate.

Comment n°29:

L105: extracted ion chromatograms of the lipophilic toxins detected from …” instead of “chromatograms of the lipophilic toxins from …”. Idem L132.

Comment n°30:

L112: It is unclear; it seems that the two methods were not applied successively. Therefore, the results presented for dietary supplements were obtained from method 1 extracts or from method 2 extracts? Clarify and justify your choice here or in material and methods.

Comment n°31:

L118: you can discard “calculated, being”.

Comment n°32:

L122: why do you use italic style for “produced” and “and”?

Comment n°33:

L123: extended seems not to be the correct word.

Comment n°34:

L129: the signal suppression is not 97.5% but only -2.5%. 97.5% correspond to the matrix effect recovery. If matrix effects where studied for the other toxins (AZA-2, PTX-2, PnTX-G) in molluscs, please also indicate your results.

Comment n°35:

L132: why the colors of the transitions are different: green/blue for A and B, and green/red for C. Please harmonize the colors whether the transitions followed are the same.

Comment n°36:

L132: why do you use italic style for “green lipped mussel powder”?

Comment n°37:

L135 and L202-206: for samples from the same brand, could you indicate if all come from the same batch or not. If you know the batch number, it could be interesting to include them in the article.

Comment n°38:

L138: in Figure 5, could you merge the two plots to obtain just one plot with the two regression represented together?

Comment n°39:

L144: As you know, LC-MS/MS methods are able to handle hundreds of compounds. You probably refer to this specific method implemented. Consequently, I cannot agree with the sentence: “LC-MS/MS methods are based on a targeted screening that only seeks to find a short list of predetermined compounds, while missing all other toxins that could be present in the sample”. In this case, it was possible and you explain that finally you added the transitions for a wide range of CIs. Therefore, you should not generalize and maybe explain that initially this specific method did not include transitions for a wide range of CIs. Please, modify accordingly.

Comment n°40:

L157: PnTXs and not PnTX.

Comment n°41:

L161: PnTX-G and not PnTX G.

Comment n°42:

L163: you said: “It seems that the maximum PnTX-G levels in the South of Europe are during winter”. Arnich et al. (2020) showed that PnTX-G peaks were observed between June and September from molluscs collected from the French Mediterranean coast (887 in 2013, 918 in 2014, 1143 in 2015, 600 in 2016 and 640 in 2017, expressed in µg/kg of total meat). Please check this sentence by considering all the existing references.

Comment n°43:

L178: “This is the first report of an emerging marine toxin is found in mussel-based supplements”

Comment n°44:

L190: not fully, agree with this sentence. You probably refer to unknown compounds. Emerging compounds or non-regulated compounds may be detected and identified with current detection methods whether the transitions are known. Modify accordingly.

Comment n°45:

L192: “To be fully…” please clarify this sentence and mention references.

Comment n°46:

L228: you said: “Mollusc samples were extracted following EU-Harmonized Standard Operating Procedure”. This SOP mention in Annex C, that water have to be added to cooked (streamed) mussels. Please indicate that this annex was not applied for those specific samples.

Comment n°47:

L238: what is mentioned in the paper as a water/methanol extraction is in fact simply a methanol extraction of a rehydrated food supplements. This has to be considered.

Comment n°48:

L242: to a volumetric flask.

Comment n°49:

L259: mention the city and country of the company as for Agilent.

Comment n°50:

L262-L267: indicate the qualifier and quantification transitions used.

Comment n°51:

L269: please mention the Table number instead of “are described below”

Comment n°52:

L270 - Table 3: the non-regulated marine toxins listed in Table 3 are not in agreement with those described in L94. If some of these toxins were not monitored in the present study, please, remove them from the table.

Comment n°53:

L273: the SOP is not a validation guideline to establish method performances; therefore, the term “according” seems not to be appropriate.  The SOP does not specify that analytical method performance assessment has to be done for three days. Could you also modify the link in reference? Currently it does not work.

Comment n°54:

L278: “Sensitivity of the method was evaluated as the slope of the calibration curve”. Could you describe this approach and mention the LOD and LOQ for the toxins monitored.

Comment n°55:

L278: “… the deviation of the curve between sample sets”. It is the samples that were between the calibration curves and not the contrary.

Comment n°56:

L 292: the instructions for authors indicate that “If you are using Word, please use either the Microsoft Equation Editor or the MathType add-on”. An example is also presented in the Toxins template.

Comment n°57:

L302-L304-L364-L375: the page number are not correctly written.

Comment n°58:

L295 - § References: it seems that the references do not respect the ACS style. Sometimes all the authors are mention whereas sometimes, only the first (e.g. Otero,P., et al.). Moreover, I am not sure of the double parenthesis in L334.

Author Response

Comment n°1: Title: I suggest reformulating the title in a way to globalize the two experiments highlightening detection of the Cis; the current title looks like a juxtaposition of two completely distinct experiments (food supplements and shellfish).

Our answer 1: The title: “First report of 13-desmethyl spirolide C in dietary supplements of green lipped mussels (Perna canaliculus) and confirmation of pinnatoxin-G in Mytilus chilensis” was replaced by “ Detection of cyclic imine toxins in dietary supplements of green lipped mussels (Perna canaliculus) and in shellfish Mytilus chilensis”.

Comment n°2: L27, L41, L114, L119, L178, and L268: the use of the term “emerging toxins” is excessive throughout the article. In several cases, “non regulated toxins” or “newly discovered” seems more adequate. Most of the toxins were recently discovered and were not really monitored so far; therefore, it is difficult to affirm that those toxins really emerging in specific areas/locations.

Our answer 2: Thank you for the comment. We included the term “non regulated toxins” and “newly discovered” instead of emerging toxins in the new version of the manuscript.

Comment n°3: L27: « first detection » is more appropriate than « first description ».

Our answer 3: Thank you for the comment. The term was replaced.

Comment n°4: L3: several words are in italics. Is there a reason?

Our answer 4: it was a mistake, italic words were corrected.

Comment n°5: L34: define the meaning of the term: “The reference compound”.

Our answer 5: The term “reference compound” is now defined. A new paragraph is included in the introduction:

“When LC-MS methodology is used, the concentration in a sample has to be referred to a predominant toxin of the group, named the reference compound (RC) and using the Toxicity Equivalent Factors (TEFs) values for the other analogues from the same group. The RCs for lipophilic toxins are YTX, AZA-1, PTX-2 and OA. The use of TEFs requires the knowledge of the toxicity of each analogue present in a sample to link analytical data into their toxicity”.

Comment n°6: L35: please indicate where those levels shall be respected? In Europe, worldwide? Could you also add regulation references?

Our answer 6: Sentence was re-write mentioning European Union (EU) and the references are now included (L43).

Comment n°7: L37: “official toxicity determination” à “official determination” seems more appropriate.

Our answer 7: The term was corrected (L45).

Comment n°8: L39: please correct the sentence. The LC-MS/MS method could be applied since 2011 according to the legislation 15/2011.

Our answer 8: The term was corrected (L47).

Comment n°9: L48: it should be noted at some part of the article that no human intoxication linked to the exposition to CIs have been reported, so far (unless new data are available).

Our answer 9: Sentence “ No incident of human intoxication has been attributed to CI so far” was included in introduction section (L52).

Comment n°10: L49: the term ”occurrence” is more appropriate than recurrence all over the article.

Our answer 10: Thank you for the comment. The term was changed all over article.

Comment n°11: L49: CIs instead of CI.

Our answer 11: The term was corrected.

Comment n°12: L64: the statement: ”the increase of emerging toxins worldwide” needs some explanation. Is it effectively an increase or due to methodology improvements and better knowledge?

Our answer 12: The sentence was replaced by “the increase of emerging toxins detection worldwide” (L71).

Comment n°13: L74: “pectenotoxin-2” and not “pentotoxin-2”

Our answer 13: The term was corrected.

Comment n°14: L73 - Figure 1: should be introduced at the end of the paragraph after it citation.

Our answer 14: Figure 1 was now introduced at the end of the introduction section as “Figure 1 shows the structure of the main compounds representatives of each toxin group analysed in this study, YTX, AZA-1, PTX-2, OA, 13-desmethyl spirolide C (SPX-13) and pinnatoxin G (PnTX-G).” (L79).

Comment n°15: L75 - § Results: you do not talk about the results below the LOD and/or LOQ for the different toxins studied. Therefore, it is difficult to understand which toxins were really analyzed. Please add some sentences in this chapter to clarify the paper.

Our answer 15. In line 102, we explain the analyses performed and toxins analysed:

“Molluscs, Mytilus chilensis (M. chilensis), Tawera gayi (T. gayi) and Meretrix lyrate (M. lyrate) were analysed with the EU-Harmonized Standard Operating Procedure (SOP) for the determination of regulated lipophilic toxins (OA, DTX-1, DTX-2, YTX, 45 OH-YTX, HomoYTX, 45 OH-HomoYTX, PTX-1, PTX-2, AZA-1, AZA-2 and AZA-3) and including SPX-13, 13,19-didesmethyl spirolide C (SPX-13,19), 20-methyl spirolide C (SPX-20G) and PnTX-G”.

A new paragraph with the LODs and LOQs were included in the new version of the manuscript (L114), although the LOQs were already included in table 1 description: “The limits of detection (LODs) were 0.1 µg/kg for OA, DTX-1, DTX-2, PTX-1, and PTX-2; 0.3 µg/kg for AZA-1, AZA-2 and AZA-3; 1.2 µg/kg for YTXs and 0.1 µg/kg for SPX-13, SPX-13,19 and PnTX-G. The limits of quantification (LOQs) were: 0.3 μg/kg (OA, DTXs and PTXs), 0.9 μg/kg (AZAs), 3.6 μg/kg (YTXs), 0.3 μg/kg (SPXs) and 0.4 μg/kg (PnTX-G).”.

Comment n°16: L78 and L92: please, replace “LC-MS reference method” by “EU-Harmonized Standard Operating Procedure (SOP)” as in L228.

Our answer 16: Thank you. Terms were corrected.

Comment n°17: L79 to L87: please, move this paragraph in the material and method part.

Our answer 17: This paragraph includes the figure 2 explanations and authors thinks it is more convenient leave this paragraph in results section near the figure.

Comment n°18: L94: YTX and not YTXs

Our answer 18: The term was corrected.

Comment n°19: L80: “PnTX-G (C42H63NO7) eluted in the minute 3 and the chromatographic peak show a high intensity of 1.2 x 107 cps “. This information are not useful.

Our answer 19: This information was deleted

Comment n°20: Figure 2, 3 and 4: sizes shall be reduced.

Our answer 20: Figures sizes were reduced.

Comment n°21: L91: “mollusks” here, whereas “molluscs” is written elsewhere. Please harmonize.

Our answer 21: Term mollusk was replace by mollusc.

Comment n°22: L92: replace “for the determination of lipophilic toxins” by “for the determination of regulated lipophilic toxins”

Our answer 22: Thank you. The sentence was replaced.

Comment n°23: L94: for SPX-13 and PnTX-G the full name is not used whereas for the two others spirolides you mention the full name and the abbreviation.

Our answer 23: Because the full name of SPX-13 and PnTX-G had been already given in the introduction (L80).

Comment n°24: L96: you used in two lines “shown”, “shows”, “showed”. Please avoid these repetitions.

Our answer 24: Thank you, other synonyms were included in the paragraph: “…are indicated in Material and method section. Figure 3 shows the chromatograms from the analysis of lipophilic marine toxins. Results evidenced the presence of ….”

Comment n°25: L97-98: there is a lack of coherence between the text and Figure 3 and Table 1 as regards the brands of M. Chilensis. In the main text, you speak about brand D and E but in Figure 1 and Table 1 there is only Brand A and B.

Our answer 25: The referee is right. It was a typographical mistake in the text description. Brand A and brand B was replaced by brand D and brand E in the text. The figures are ok.

Comment n°26: L 98: « Besides » ?

Our answer 26: Term was replaced by also.

Comment n°27: L 100: “Table 1 shows the quantification” is not a complete sentence

Our answer 27: Sentence was rewrite: “Table 1 collects the quantification for each toxin present in the samples upper the LOQs”.

Comment n°28: L 101: can you mention the years instead of “recently”? It will be more accurate.

Our answer 28: yes, no problem. Years were included. “PnTXs, produced by Vulcanodinium rugosum dinoflagellate, have been detected in seafood from the Mediterranean Sea in 2013 and 2018 and the Atlantic coast of Spain last year”. And the references are also in the text.

Comment n°29: L105: extracted ion chromatograms of the lipophilic toxins detected from …” instead of “chromatograms of the lipophilic toxins from …”. Idem L132.

Our answer 29: We reformulated the sentences. The chromatograms were obtained in MRM mode. So, we included the term MRM (multiple reaction monitored) instead of “extracted ion chromatograms”.

Comment n°30: L112: It is unclear; it seems that the two methods were not applied successively. Therefore, the results presented for dietary supplements were obtained from method 1 extracts or from method 2 extracts? Clarify and justify your choice here or in material and methods.

Our answer 30: Sorry for the confusion. The paragraph was clarified and justified. “Using both extraction methods, results showed the existence of the emerging toxin SPX-13 in 6 out 9 samples analysed after both extractions and no other lipophilic toxins were found. Comparing extraction methods, better results were obtained for the extraction 2”. The chromatograms showed in the manuscript correspond to extraction method 2, and now are indicated in the new version of the manuscript.

Comment n°31: L118: you can discard “calculated, being”.

Our answer 31: Words were deleted.

Comment n°32: L122: why do you use italic style for “produced” and “and”?

Our answer 32: It was a typographical error. The italic style was removed.

Comment n°33: L123: extended seems not to be the correct word.

Our answer 33: “extended” was replace by “common”

Comment n°34: L129: the signal suppression is not 97.5% but only -2.5%. 97.5% correspond to the matrix effect recovery. If matrix effects where studied for the other toxins (AZA-2, PTX-2, PnTX-G) in molluscs, please also indicate your results.

Our answer 34: The sentence was rewrite: “ The signal suppression/ enhancement (SSE) value due to the matrix was 97.5 %, so there was a negative effect which entailed a suppression of signal (-2.5%)”.

In material and methods, we had included that the signal suppression/ enhancement (SSE) value due to matrix was calculated according to the following equation:

SSE (%) = 100 x (Slope of spiked extract curve /Slope of standards curve in solvent).

If SSE value is equal to 100%, no matrix effect is observed whereas if the value is higher than 100% means a positive matrix effect due to an enhancement of the ionization. If this value is less than 100% there is a negative effect, which entails a suppression of the signal.

A new paragraph with the matrix effects for the other toxins were included: “The signal suppression/ enhancement (SSE) value due to the matrix was 94.2%, 40.5% and 65.0% for PTX-2, AZA-2 and PnTX-G, respectively. For three toxins, there were a negative effect which entailed a suppression of signal” (L120).

Comment n°35: L132: why the colors of the transitions are different: green/blue for A and B, and green/red for C. Please harmonize the colors whether the transitions followed are the same.

Our answer 35: Yes, the transitions are the same, but the colors are given by the LC-MS/MS software. To avoid confusion, we have indicated the transitions in each figure in the new version of the manuscript.

Comment n°36: L132: why do you use italic style for “green lipped mussel powder”?

Our answer 36: Italic style was removed.

Comment n°37: L135 and L202-206: for samples from the same brand, could you indicate if all come from the same batch or not. If you know the batch number, it could be interesting to include them in the article.

Our answer 37: Yes, the batch number is in the product. Samples from the same brand come from the same batch and this information is now included in material and methods

Comment n°38: L138: in Figure 5, could you merge the two plots to obtain just one plot with the two regression represented together?

Our answer 38: No, we cannot make it. The two regressions are represented separately since they are taken from the software as two different graphs.

Comment n°39: L144: As you know, LC-MS/MS methods are able to handle hundreds of compounds. You probably refer to this specific method implemented. Consequently, I cannot agree with the sentence: “LC-MS/MS methods are based on a targeted screening that only seeks to find a short list of predetermined compounds, while missing all other toxins that could be present in the sample”. In this case, it was possible and you explain that finally you added the transitions for a wide range of CIs. Therefore, you should not generalize and maybe explain that initially this specific method did not include transitions for a wide range of CIs. Please, modify accordingly.

Our answer 39: When we mentioned “LC-MS/MS methods are based on a targeted screening that only seeks to find a short list of predetermined compounds, while missing all other toxins that could be present in the sample”, we referred to other unknown toxins. So, the sentence was re-writen: “LC-MS/MS methods are based on a targeted screening that seeks to find a list of know compounds, while missing other unknown toxic compounds that could be present in the sample” (L175).

Comment n°40: L157: PnTXs and not PnTX.

Our answer 40: Term was corrected

Comment n°41: L161: PnTX-G and not PnTX G.

Our answer 41: Term was corrected.

Comment n°42: L163: you said: “It seems that the maximum PnTX-G levels in the South of Europe are during winter”. Arnich et al. (2020) showed that PnTX-G peaks were observed between June and September from molluscs collected from the French Mediterranean coast (887 in 2013, 918 in 2014, 1143 in 2015, 600 in 2016 and 640 in 2017, expressed in µg/kg of total meat). Please check this sentence by considering all the existing references.

Our answer nº 42: Very interesting. Thank you for the information. Both references were included:

“Lamas et al. (2019) found that the maximum PnTX-G levels in the South of Europe are during winter. However, Arnich et al. (2020) showed that PnTX-G peaks were observed between June and September from molluscs collected from the French Mediterranean coast between 2013 and 2017”

Comment n°43: L178: “This is the first report of an emerging marine toxin is found in mussel-based supplements”

Our answer nº 43: Term “is” was deleted.

Comment n°44: L190: not fully, agree with this sentence. You probably refer to unknown compounds. Emerging compounds or non-regulated compounds may be detected and identified with current detection methods whether the transitions are known. Modify accordingly.

Our answer nº 44: We totally agree. Authors had referred to unknown compounds, so the word emerging was replaced by unknown.

Comment n°45: L192: “To be fully…” please clarify this sentence and mention references.

Our answer nº 45: There is not refence because it is our opinion at the end of the discussion section. Despite this, we have removed “To be fully transparent with respect to consumers’ safety” and the sentence was rewrite as: “To guarantee the consumers´s safety, food quality assurance would have to detect the presence of toxic unknow compounds which include newly discovered toxins, the detection of know toxins in areas and species where they had not been previously recorded”.

Comment n°46: L228: you said: “Mollusc samples were extracted following EU-Harmonized Standard Operating Procedure”. This SOP mention in Annex C, that water have to be added to cooked (streamed) mussels. Please indicate that this annex was not applied for those specific samples.

Our answer nº 46: yes, it was not applied. In the new version of the manuscript we have included the following sentence: “All bivalves were extracted according the procedure from the section 6.2 of the SOP, the Annex C was not applied for cooked samples”. So, the text in the manuscript is as follows:

“Mollusc were extracted following the EU-Harmonised Standard Operating Procedure (SOP) for determination of lipophilic marine biotoxins in molluscs by LC-MS/MS. All bivalves were extracted according the procedure from section 6.2 of the SOP, the Annex C was not applied for cooked samples” (L311).

Comment n°47: L238: what is mentioned in the paper as a water/methanol extraction is in fact simply a methanol extraction of a rehydrated food supplements. This has to be considered.

Our answer nº 47: yes, it is the same. For authors is fine as per the referee prefer, so two solvents were removed and the sentence was re-write as follows: “Food supplements were extracted adding methanol to the powder samples (extraction 1) or using methanol after a rehydration of the food supplements (extraction 2) (L321).

Comment n°48: L242: to a volumetric flask.

Our answer nº 48: The letter “a” was added.

Comment n°49: L259: mention the city and country of the company as for Agilent.

Our answer nº 49: City and country were included (L343).

Comment n°50: L262-L267: indicate the qualifier and quantification transitions used.

Our answer nº 50: Qualifier and quantification transitions are now indicated: “Analysis were performed in multiple reaction monitoring (MRM) acquisition mode, selecting two transitions per molecule. The former transition of each toxin indicated below was used for quantification and second one as qualifier” (L347).

Comment n°51: L269: please mention the Table number instead of “are described below”

Our answer nº 51: the term was replaced by table 3.

Comment n°52: L270 - Table 3: the non-regulated marine toxins listed in Table 3 are not in agreement with those described in L94. If some of these toxins were not monitored in the present study, please, remove them from the table.

Our answer nº 52: All toxins from table 3 were analysed, so authors did not remove them from the table.

In Line 103, authors indicated that first, samples were analysed for all regulated lipophilic marine toxins plus SPX-13, SPX-13,19, SPX-20 and PnTX-G (these are imine cyclic from which standards are available).

Then, in line 270, authors indicate that “due to CIs include a long list of compounds and considering the recurrence of PnTX-G worldwide and the existence of SPX-13 in the food supplements, we also considered the possible existence of other CI in the samples. Thus, dietary supplements and molluscs were reanalysed using a MS method included a wide range of CI for which standards are not available”. Transitions are included in table 3 and this is indicated in the text.

Comment n°53: L273: the SOP is not a validation guideline to establish method performances; therefore, the term “according” seems not to be appropriate. The SOP does not specify that analytical method performance assessment has to be done for three days. Could you also modify the link in reference? Currently it does not work.

Our answer nº 53: Thank you for the comment. “According” was removed. Sentence was rewrite as follows: “Analytical method performance assessment was performed for three days and following the EU-Harmonized SOP for Lipophilic toxins and the guidelines proposed by the Regulation (EC) 657/2002” (L360).

The link is working for us: http://www.aecosan.msssi.gob.es/AECOSAN/docs/documentos/laboratorios/LNRBM/ARCHIVO2EU-Harmonised-SOP-LIPO-LCMSMS_Version5.pdf

Comment n°54: L278: “Sensitivity of the method was evaluated as the slope of the calibration curve”. Could you describe this approach and mention the LOD and LOQ for the toxins monitored.

Our answer nº 54: The following sentence is included in the manuscript: “Sensitivity of the method was evaluated with the slope of the calibration curves and the limit of detection (LOD) and limit of quantification (LOQ) which were calculated based on an S/N > 3 and an S/N > 10, respectively, using triplicate injections (n = 3) of standard solutions with concentrations near the limits”.

The bigger slope, the greater the signal for the same concentration and the better the sensitivity. LODs and LOQs are indicated in the text as follows:

“Limits of detections (LODs) were 0.1 µg/kg for OA, DTX-1, DTX-2, PTX-1, and PTX-2; 0.3 µg/kg for AZA-1, AZA-2 and AZA-3; 1.2 µg/kg for YTXs and 0.1 µg/kg for SPX-13, SPX-13,19 and PnTX-G. Table 1 shows the toxin quantification in the samples. Levels were low, up to 4.00 µg/kg, 4.33 µg/kg and 10.88 µg/kg for PnTX-G, AZA-2 and PTX-2, respectively. For these toxins LOQ were: 0.3μg/kg (PTX-2), 0.9 μg/kg (AZA-2) and 0.4μg/kg (PnTX-G)”.

Comment n°55: L278: “… the deviation of the curve between sample sets”. It is the samples that were between the calibration curves and not the contrary.

Our answer nº 55: The sentence was rewrite. We meant “the slope variation between the sets of the calibration curve had to be lower than 25% to be considered as acceptable” (L364).

Comment n°56: L 292: the instructions for authors indicate that “If you are using Word, please use either the Microsoft Equation Editor or the MathType add-on”. An example is also presented in the Toxins template.

Our answer nº 56: Thank you. Microsoft Equation Editor was used to rewrite the equation.

Comment n°57: L302-L304-L364-L375: the page number are not correctly written.

Our answer nº 57: the page numbers were corrected in the 4 references.

Comment n°58: L295 - § References: it seems that the references do not respect the ACS style. Sometimes all the authors are mention whereas sometimes, only the first (e.g. Otero,P., et al.). Moreover, I am not sure of the double parenthesis in L334.

Our answer nº 58: The referee is right. All references were checked and corrected according to the journal stile.

Reviewer 2 Report

The present manuscript describes the identification of several marine lipophilic toxins in non-EU seafood and seafood-based food supplements.

The results are clearly described and the experimental design sufficiently proper for the aim of the study. However, the manuscript lack of a suitable discussion (and a scanty literature citation) that must be heavily improved before the manuscript becomes suitable for its publication in Toxins. In particular, discussion of the results should focus on safety aspects, considering also the toxicology-oriented readership of the journal.

Major points:

  1. Regarding azaspiracids, it is not clear if authors included all the regulated analogues (AZA-1, -2 and -3) and they found contamination only by AZA-2 or if they performed the analysis only for AZA-2. This should be better explained in the results section.
  2. Accordingly to the previous point, the fact that only AZA-2 was found is a very curious fact (if this is the case). If authors did not find any traces of the other analogues (in particular AZA-1, the most potent under a toxicological point of view) this should be discussed, also in view of safety issues. New studies on azaspiracids TEFs were recently published: does the level of AZA-2 contamination could represent a toxicological problem? Can the authors put their results in relation to the available toxicological data? At least, these studies should be mentioned.
  3. Similarly to the previous point, the discussion should be improved also for pinnatoxin-G data. Very recently new toxicological evaluations have been published (also in this case, the relevant study was not cited): authors should compare the amount of toxin identified in mussels with toxicity data to discuss if these contamination could represent a safety issue for consumers. Again, literature citation (also in this case regarding toxicological data) should be implemented.
  4. For DSP toxins and yessotoxin, again, it is not clear if they were included in the analysis without finding any traces. I cannot find any info in the results, but from the method section I guess they were checked. Authors should state in the results that they did not find any trace (or the quantitation was < LOQ?). Authors should implement this part of the results to be more clear.
  5. In some case, the same specimen is contaminated by more than one toxin, such as in the case of Tawera Gayi (AZA-2 and PTX-2). Which are the toxicological implications? Are some toxicological studies already published reporting the effects of this co-association?

Minor points:

  1. Explain PnTXs acronym in the abstract
  2. In the abstract, the sentence at lines 20-21 is misleading. As far as I know, marine toxins have been already reported in food supplements.
  3. Introduction, line 32: legislated where? In EU? I think something has been changed for yessotoxins. This should be checked and modified.
  4. Authors should explain the ratio underneath the choice of the samples: why those geographical sites? Why those species?

Author Response

The present manuscript describes the identification of several marine lipophilic toxins in non-EU seafood and seafood-based food supplements.

The results are clearly described and the experimental design sufficiently proper for the aim of the study. However, the manuscript lack of a suitable discussion (and a scanty literature citation) that must be heavily improved before the manuscript becomes suitable for its publication in Toxins. In particular, discussion of the results should focus on safety aspects, considering also the toxicology-oriented readership of the journal.

Our answer: Thank you for the comment. Several paragraphs addressing the toxicology and safety aspects are now included in the manuscript (discussion section). L204-L240

Major points:

Comment 1: Regarding azaspiracids, it is not clear if authors included all the regulated analogues (AZA-1, -2 and -3) and they found contamination only by AZA-2 or if they performed the analysis only for AZA-2. This should be better explained in the results section.

Our answer 1: Exactly. First, we analysed all the regulated toxins, including AZA-1, AZA-2 and AZA-3 and only AZA-2 was found. We have indicated all the toxins analysed in line 105.

Comment 2: Accordingly to the previous point, the fact that only AZA-2 was found is a very curious fact (if this is the case). If authors did not find any traces of the other analogues (in particular AZA-1, the most potent under a toxicological point of view) this should be discussed, also in view of safety issues. New studies on azaspiracids TEFs were recently published: does the level of AZA-2 contamination could represent a toxicological problem? Can the authors put their results in relation to the available toxicological data? At least, these studies should be mentioned.

Our answer 2: Thank you for the comment. We have added references from studies in which AZA-2 was the most abundant or unique AZA analogue among three analysed (AZA-1, AZA-2 and AZA-3). The new studies on AZAs TEFs were included in the manuscript. In the present study, the AZA-2 content does not represent a toxicological problem and we have included this information in the manuscript (L204-L226).

Comment 3: Similarly, to the previous point, the discussion should be improved also for pinnatoxin-G data. Very recently new toxicological evaluations have been published (also in this case, the relevant study was not cited): authors should compare the amount of toxin identified in mussels with toxicity data to discuss if these contaminations could represent a safety issue for consumers. Again, literature citation (also in this case regarding toxicological data) should be implemented.

Our answer 3: The discussion has also been improved for pinnatoxin-G data. The very recently publications have been included in the new version of the manuscript. Amount of toxin identified in mussels was compare with toxicity data and safety issues for consumers were now included (L202).

TOXICOLOGICAL DATA OF PTXs

Comment 4: For DSP toxins and yessotoxin, again, it is not clear if they were included in the analysis without finding any traces. I cannot find any info in the results, but from the method section I guess they were checked. Authors should state in the results that they did not find any trace (or the quantitation was < LOQ?). Authors should implement this part of the results to be more clear.

Our answer 4: Yes, DSP and yessotoxins were included and it is also indicated in line 103. Trace of these compounds were not detected upper LODs. LODs were also included in the new version of the manuscript and this issue is also indicated in the manuscript (L120) “…and no other toxin was detected under LOQs.”. 

Comment 5: In some case, the same specimen is contaminated by more than one toxin, such as in the case of Tawera Gayi (AZA-2 and PTX-2). Which are the toxicological implications? Are some toxicological studies already published reporting the effects of this co-association?

Our answer 5: In Asia, several combinations of lipophilic marine toxins were reported, including OA/YTX, OA/PTX-2, YTX/OA, PTX-2/OA, PTX-2/GYM, GYM/PTX-2 and AZA-2/PTX-2. It appears that OA is the most often recorded lipophilic toxin in mixtures, as well as the predominant toxin in the mixture. With regards to the simultaneous exposure to AZA-2 and PTX-2. Toxicological studies reporting the effects of this co-association were not found in the literature.

Minor points:

Comment 6: Explain PnTXs acronym in the abstract

Our answer 6: The term pinnatoxins was included in the abstract.

Comment 7: In the abstract, the sentence at lines 20-21 is misleading. As far as I know, marine toxins have been already reported in food supplements.

Our answer 7: The terms “marine” was replaced by cyclic imine. Whole sentence: “To the best of our knowledge, this is the first time that an emerging cyclic imine toxin in dietary supplements is reported.

Comment 8: Introduction, line 32: legislated where? In EU? I think something has been changed for yessotoxins. This should be checked and modified.

Our answer 8: yes, the EU was included.

Comment 9: Authors should explain the ratio underneath the choice of the samples: why those geographical sites? Why those species?

Our answer 9: “Species (and locations) were chosen at random, since they are frequent products available in local markets”. This explanation is now included in the manuscript.

Reviewer 3 Report

Manuscript entitled "First report of 13-desmethyl spirolide C in dietary supplements of green lipped mussels (Perna canaliculus) and confirmation of pinnatoxin-G in Mytilus chilensis" presents the result of marine toxins analysis in seafood.

Authors have performed extraction and LC-MS/MS analysis of mussels, calms and food supplements based on mussels´formulations.

The methodology used allowed for the identification of toxins in certain products analysed. Fortunately, some of the toxins were not found in some products.

Methodology of extraction and LC-MS analysis seems to be correct. One question arise: Why was MS analysis performed using HPLC-pure solvents?

Fig. 3 and Fig. 4 should be described in more details. Is it MRM? What do the colours mean?

Did Author try to apply other extraction methods for the comparison?

Author Response

Question 1: One question arise: Why was MS analysis performed using HPLC-pure solvents?

Our answer 1: To avoid impurities or contaminants in solvents which could have an impact on the sensitivity of a system or interferences.

Question 2: Fig. 3 and Fig. 4 should be described in more details. Is it MRM? What do the colours mean?

Our answer 2: Figure 3 and Figure 4 was now described in more detail; they are multiple reaction monitoring (MRM) chromatograms and this information is included, not only in material and methods, but also in the legend in the new version of the manuscript. The colours belong to toxin transitions. To avoid confusion, we have now included the transitions in the figure 3 and figure 4.

Question 3: Did Author try to apply other extraction methods for the comparison?

Our answer 3: For molluscs we have used one extraction and for food supplements we have compared two extractions methods based in methanol or water: methanol. And all are indicated in the manuscript.

Reviewer 4 Report

  1. I think the title of the research is two different things. Better find a common denominator between the two compounds. The first report is so ambiguous here.
  2. Line 8, spell out EU.
  3. Southeast Pacific is vague. Specify location.
  4. Make the abstract more comprehensive. Abbreviations here cannot be understood.
  5. The first paragraph is not a good introduction, and the whole Introduction should be improved and must be relatable to the whole audience, not just Europeans.
  6. Figure 1, Make figures similar.
  7. Line 79, what is legislated?
  8. Figure 2, draw the structure of fragment ions. Also, what is the solvent condition here?
  9. I think in lines 91 and 92, abbreviations make the readers dizzy. Is this common? Or better spell out.
  10. I cannot understand Figure 3.
  11. Table 1, make it more informative. What are brands A to C? What is ⁓?
  12. Tables 1 and 2 can be combined.
  13. Line 140, what is CI?
  14. What is the novelty of your study? It was already known, and making it as a first report does not make it novel. What are the possible ways or accumulation of 13-desmethyl in P. canaliculus? Also, the same as the other compound.

Author Response

Comment 1: I think the title of the research is two different things. Better find a common denominator between the two compounds. The first report is so ambiguous here.

Our answer 1: Thank you for the suggestion. The title: “First report of 13-desmethyl spirolide C in dietary supplements of green lipped mussels (Perna canaliculus) and confirmation of pinnatoxin-G in Mytilus chilensis” was replaced by “ Detection of cyclic imine toxins in dietary supplements of green lipped mussels (Perna canaliculus) and in shellfish Mytilus chilensis”.

Comment 2: Line 8, spell out EU.

Our answer 2: “European Union” was included.

Comment 3: Southeast Pacific is vague. Specify location.

Our answer 3: We do not have more information since only this is included in the product label. We cannot specify more.

Comment 4: Make the abstract more comprehensive. Abbreviations here cannot be understood.

Our answer 4: Thank you for the comment. All abbreviations were deleted and now the abstract is more comprehensive.

Comment 5: The first paragraph is not a good introduction, and the whole Introduction should be improved and must be relatable to the whole audience, not just Europeans.

Our answer 5: The first paragraph was re-rewrite and all introduction is now focus to whole audience.

Comment 6: Figure 1, Make figures similar.

Our answer 6: Yes, thank you. The structures were homogenised and made them like the other ones.

Comment 7: Line 79, what is legislated?

Our answer 7: the toxin legislated are in the introduction. L34.

“Only few compounds of each group are regulated: yessotoxin (YTX), homo-yessotoxin (Homo-YTX), 45- hydroxy-yessotoxin (45-OH-YTX), 45- hydroxy-homo-yessotoxin (45 -OH-homo-YTX), azaspiracid-1 (AZA-1), azaspiracid-2 (AZA-2), azaspiracid-3 (AZA-3), pectenotoxin-1 (PTX-1), pectenotoxin-2 (PTX-2), OA, dinophysistoxin-1 (DTX-1), dinophysistoxin-2 (DTX-2) and dinophysistoxin-3 (DTX-3)”.

Comment 8: Figure 2, draw the structure of fragment ions. Also, what is the solvent condition here?

Our answer 8: Thank you for your suggestion, however, we do not find a useful thing to draw the structure of the ions since they are already published in Moreiras et a. 2020, International Journal of Environmental Research and Public Health, 17 (1), 281https://doi.org/10.3390/ijerph17010281. However, we have included this reference in the manuscript.

Comment 9: I think in lines 91 and 92, abbreviations make the readers dizzy. Is this common? Or better spell out.

Our answer 9: We think it is better to spell out the species since it is the first time they are mentioned in the text.

Comment 10: I cannot understand Figure 3.

Our answer 10: The figure 3 is the Multiple reaction monitoring (MRM) chromatogram of Mytilus chilensis (M. chilensis) brand A and band B and Tawera gayi brand C. The sentence was rewritten, and MRM are now included.

Comment 11: Table 1, make it more informative. What are brands A to C? What is ~?

Our answer 11: The term ~ means below the limit of quantitation (LOQ) and it was already explained in the table description. The terms “Brands A, B and C” are included in the table to evidence 3 different commercial brands from market, but obviously, we cannot provide the name of the brands.

Comment 12: Tables 1 and 2 can be combined.

Our answer 12: Thank you for the suggestion, but authors think that it is more useful to have the tables separately near their description in the manuscript.

Comment 13: Line 140, what is CI?

Our answer 13: Cyclic imines and it was already explained and abbreviated in the introduction part.

Comment 14: What is the novelty of your study? It was already known, and making it as a first report does not make it novel. What are the possible ways or accumulation of 13-desmethyl in P. canaliculus? Also, the same as the other compound.

Our answer 14: The possible ways or accumulation of 13-desmethyl spirolide C (SPX-13) in P. canaliculus is not comprised in the aim of the present study and we haven’t studied it. But it would be an interesting thing for futures investigations or collaborations, so, thank you for the comment. The novelty of this study is to find spirolides in food supplements of green lipped mussels’ powder and pinnatoxins in Chilean Mussels. With this, we are identifying new matrix and locations where the emerging toxins are occurring, and we are contributing with information about new possible risks in these food products. These toxins are not legislated. Moreover, we provide an adequate extraction procedure with high toxin recovery (97%) and able to quantify SPX-13 in the range of 13.4 - 427.35 µg/kg with almost inexistent matrix effect.

Round 2

Reviewer 2 Report

The manuscript has significantly improved after the revision process; thus, it is suitable for publication in its present form.

I suggest to modify the term Toxicity Equivalent Factor in Toxicity Equivalency Factor, at line 39.

Author Response

Comment 1. The manuscript has significantly improved after the revision process; thus, it is suitable for publication in its present form.

Our answer: We are grateful for the positive comment of the reviewer. We have revised the manuscript to improve the understanding of all work performed.

Comment 2. I suggest to modify the term Toxicity Equivalent Factor in Toxicity Equivalency Factor, at line 39.

Our answer: We agree with the reviewer. The term “Toxicity Equivalent Factor” was replace by “Toxicity Equivalency Factor” (line 39).

Reviewer 4 Report

The manuscript entitled, Title: First report of 13-desmethyl spirolide C in dietary supplements of green-lipped mussels (Perna canaliculus) and confirmation of pinnatoxin-G in Mytilus chilensis, needs major revisons before it could be published for Toxins.

The following points should be considered.

  1. Line 8, what is novel risk?
  2. Line 124, what is por? I suggest to have English correction.
  3. Figure 3, the values in the axes cannot be seen.
  4. What is your matrix in Figure 5?
  5. Lines 171 to 174. This is a weak statement.
  6. Table 4. Make this table more informative.
  7. Why Table 4 precedes Table 3?
  8. Table 3. You only optimize CE? How about other parameters?
  9. Line 368, the more samples the better. It should be n = 5. If you can increase this, the better.

Author Response

The following points should be considered.

Our answer: The authors are very grateful for the time invested and the comments suggested by the reviewer. We have taken the suggestions into account and hope that the manuscript is clearer. All changes performed in the new version of the manuscript are green underline.

Question 1: Line 8, what is novel risk?

Our answer 1: The review is asking about this paragraph: “Seafood represents a significant part of the human staple diet. In the recent years, the identification of emerging lipophilic marine toxins has increased, leading to the potential for consumers to be intoxicated by these novel risks”. The term “novel risks” was replaced by “toxins”.

Question 2: Line 124, what is por? I suggest to have English correction.

Our answer 2: Thank you for the suggestion. The term “por” was replaced by “for”. We have reviewed all the manuscript and we hope that the new version of the manuscript is more fluid.

Question 3: Figure 3, the values in the axes cannot be seen.

Our answer 3: We agree with the reviewer. The size of Figure 3 was increased.

Question 4: What is your matrix in Figure 5?

Our answer 4: The matrix in Figure 5 is the powder from the dietary supplements of green lipped mussels. This information has been indicated in Figure 5 legend in this new version of the manuscript: “Figure 5: Calibration curves for spirolide 13-desmethyl spirolide C (SPX-13) in methanol (A) and in matrix (powder from the dietary supplements of green lipped mussels) (B) in the range 0.19–25 ng/mL.”

Question 5: Lines 171 to 174. This is a weak statement.

Our answer 5: The comment reviewer is about this paragraph:

“Due to CIs consist of a large number of molecules, mainly without commercially available standards, and considering the occurrence of PnTX-G worldwide and the existence of SPX-13 in the food supplements, we also considered the possible existence of other CIs in the samples. Moreover, it seems probable that PnTX-G is the precursor of PnTX-A, PnTX-B and PnTX-C due to metabolic and hydrolytic transformations in molluscs”.

We meant that LC-MS/MS methods are based on a targeted screening that seeks to find a list of known compounds, while missing other unknown toxic compounds that could be present in the sample.

The paragraph was rewrite as follows: “Due to CIs consist of a large number of molecules, mainly without commercially available standards, and considering the occurrence of different analogues of PnTXs worldwide and SPX-13 in the food supplements, we also considered the possible existence of other CIs different from those first analysed in the samples. LC-MS/MS methods are based on a targeted screening that seeks to find a list of known compounds, while missing other unknown toxic compounds that could be present in the sample”.

Question 6: Table 4. Make this table more informative.

Our answer 6: The table 4 (called table 3 in the new version of the manuscript) includes a total of 21 marine food products purchased during November and December 2019. Product type, origin and amount obtained from the market are described in table 3. They were from 4 locations (New Zealand, Chile, SouthEast Pacific and Vietnam) and belonged to 6 commercial brands (called A, B, C, D, E and F) (included also in the table). Samples 1-6 are frozen mussels (M. chilensis) and samples 6-12 are frozen clams (T. gayi and M. lyrate), all purchased in a local market in Lugo (Spain). Samples 13-21 are green lipped mussels’ powder (P. canaliculus) and they were obtained by 3 different EU distributors (included in the table). Samples from the same brand come from the same batch and species (and locations) were chosen at random, since they are frequent products available in local markets.

All this information is given in the table and in the text. Moreover, the terms “Brands A, B and C” are included in the table to evidence 3 different commercial brands from market. If there is something else the review would like to know about the table 4, we would be happy to give the information, but all the information available about the products is included in the manuscript, to be honest we do not have more information..

Question 7: Why Table 4 precedes Table 3?

Our answer 7: We agree with the review. It was a typographical mistake. Numbers in the table legend were inverted and now number 3 precedes 4.

Question 8: Table 3. You only optimize CE? How about other parameters?

Our answer 8: No, we have optimised more parameters. We have added three new columns with the parameters: fragmentor voltage (Frag) values, cell acceleration voltage (CAV) values and polarity for each emerging toxin (table 4). Some of this information (CAV and polarity) was already included in the manuscript. However, we have included this information in the table as reviewer requested together with the fragmentor voltage values.

Question 9: Line 368, the more samples the better. It should be n = 5. If you can increase this, the better.

Our answer 9: Thank you for the comment. We agree with the review that n=5 is better than n=3, but we performed our experiment with triplicate injections as many other studies provided in the bibliography.

Round 3

Reviewer 4 Report

1. Table 3 is redundant.The authors can pool cooked mussels and just write brands A and B, and n=3?, and  do the rest to the food products.

2. Same as in Table 4. Please delete polarity and CAV and write as  Note. What is Frag? 

Author Response

Please find attached a new version of the manuscript. We have considered the review suggestions and the changes are indicated in yellow underline.

Comments and Suggestions for Authors

  1. Table 3 is redundant. The authors can pool cooked mussels and just write brands A and B, and n=3?, and  do the rest to the food products.

Our answer 1: No, because it is not n=3. They are 3 different products buying by separately. All food products were buying by separately. And we want to highlight this.

  1. Same as in Table 4. Please delete polarity and CAV and write as  Note. What is Frag?

Our answer 2: polarity and CAV was removed from table 4 and the information was included in the text as follows:

“The MS/MS method for the screening of emerging toxins are described in table 4. The MS/MS operated in positive ionization mode and the cell acceleration voltage (CAV) was 4 volts. Dwell was 6 for all toxins”.

Frag is fragmentor voltage an it is already abbreviated in the legend.